# Stem Cell Therapy for Neonatal Hypoxic-Ischemic Encephalopathy: A Systematic Review of Preclinical Studies

**DOI:** 10.3390/ijms22063142

**Published:** 2021-03-19

**Authors:** Inês Serrenho, Miguel Rosado, Alexandra Dinis, Carla M. Cardoso, Mário Grãos, Bruno Manadas, Graça Baltazar

**Affiliations:** 1Centro de Investigação em Ciências da Saúde (CICS-UBI), Universidade da Beira Interior, 6200-506 Covilhã, Portugal; inesserrenho2@gmail.com; 2CNC—Center for Neuroscience and Cell Biology, University of Coimbra, 3004-504 Coimbra, Portugal; mmva.rosado@gmail.com (M.R.); mgraos@biocant.pt (M.G.); 3Pediatric Intensive Care Unit, Hospital Pediátrico, Centro Hospitalar e Universitário de Coimbra, 3000-075 Coimbra, Portugal; alexandrasdinis@gmail.com; 4Crioestaminal, 3060-197 Cantanhede, Portugal; carla.cardoso@crioestaminal.pt; 5Institute for Interdisciplinary Research, University of Coimbra (IIIUC), 3030-789 Coimbra, Portugal; 6Biocant, Technology Transfer Association, 3060-197 Cantanhede, Portugal

**Keywords:** hypoxic-ischemic encephalopathy, stem cell therapy, umbilical cord blood cells, umbilical cord tissue, mesenchymal stem/stromal cells, therapeutic hypothermia

## Abstract

Neonatal hypoxic-ischemic encephalopathy (HIE) is an important cause of mortality and morbidity in the perinatal period. This condition results from a period of ischemia and hypoxia to the brain of neonates, leading to several disorders that profoundly affect the daily life of patients and their families. Currently, therapeutic hypothermia (TH) is the standard of care in developing countries; however, TH is not always effective, especially in severe cases of HIE. Addressing this concern, several preclinical studies assessed the potential of stem cell therapy (SCT) for HIE. With this systematic review, we gathered information included in 58 preclinical studies from the last decade, focusing on the ones using stem cells isolated from the umbilical cord blood, umbilical cord tissue, placenta, and bone marrow. Outstandingly, about 80% of these studies reported a significant improvement of cognitive and/or sensorimotor function, as well as decreased brain damage. These results show the potential of SCT for HIE and the possibility of this therapy, in combination with TH, becoming the next therapeutic approach for HIE. Nonetheless, few preclinical studies assessed the combination of TH and SCT for HIE, and the existent studies show some contradictory results, revealing the need to further explore this line of research.

## 1. Introduction

Hypoxic-ischemic encephalopathy (HIE) is one of the major causes of neonatal death and long-term disability, leading to chronic motor and cognitive impairments [1]. Several disorders are associated with HIE, namely epilepsy, cerebral palsy [2], attention deficit hyperactivity disorder, seizures, hearing and vision loss, language disorders, and cognitive delay. The different outcomes of this condition can be severe, profoundly affecting the daily life of patients and their families. This condition also represents a major economic burden to the government and caretakers [3].

About a quarter of neonatal deaths worldwide can be attributed to perinatal asphyxia [4]. The estimated incidence of HIE is variable across studies, ranging from 1 to 8 cases per 1000 live births [5]. In developed countries, neonatal HIE incidence is approximately 0.5 to 1 case per 1000 live births [6]; however, the global estimate is highly influenced by the higher incidence found in developing countries [7]. Infants diagnosed with HIE have a reserved prognosis since the HIE mortality rate is about 25%, and 20% of the survivors develop moderate to severe long-term impairment [7].

Neonatal HIE is originated from an insult that involves a period of reduced blood flow and oxygen delivery to the brain of neonates—ischemia and hypoxia, respectively. This hypoxic-ischemic (HI) event can occur due to placental abruption, uterine rupture, and cord prolapse, among others [5]. Research shows that this type of injury comprises different stages: energy depletion, inflammation, excitotoxicity, oxidative stress, and apoptosis [8]. Months after the HI insult, alterations caused by this injury are still occurring, namely late cell death, remodeling of the injured brain, astrogliosis, as well as epigenetic changes [1]. Magnetic resonance imaging studies of term newborns diagnosed with HIE revealed characteristic patterns of brain injury that can relate to mechanisms, severity, and duration of the HI insult, such as the parasagittal cerebral injury pattern or watershed injury, involving cortical gray matter and subcortical white matter, which can be related with cerebral hypoperfusion and low sustained systemic blood pressure, and the selective neuronal necrosis pattern either involving basal ganglia and brain stem in severe acute events or in a diffuse global injury when severe but also prolonged HI events occur [9].

Due to poor antioxidant defenses and higher fatty acid concentrations, the developing brain is more susceptible to oxidative stress, therefore being highly vulnerable to hypoxic-ischemic insults [10]. The HI insult affects differently the preterm and term brain, originating different types of injury [11,12].

One of the most widely used animal models for the assessment of hypoxic-ischemic brain damage in the neonatal brain is the Rice–Vannucci (RV) murine model [1,13]. The RV model was first described in postnatal day-seven (P7) rats, and the protocol consists of the unilateral permanent occlusion of the common carotid artery (CCA)—ischemia—followed by exposure to a variable period of reduced oxygen levels (usually 8%)—hypoxia. The degree of brain damage severity is highly dependent on both the duration of the hypoxic period [14] and the arteries occluded, since the ligation of both the common and external carotid arteries, instead of only CCA ligation, results in a larger and more consistent brain lesion [15]. Concerning the rodent age when the HI insult is inflicted, the majority of the studies use P7 rodents, which present a brain development equivalent to the human fetus at 32–34 weeks of gestation, while other use P10 rodents, which present a brain development equivalent to the human newborn at term [16].

## 2. Therapeutic Hypothermia—The Standard of Care

As HIE is an evolving process, the timing of the treatment is crucial. Other than supportive intensive care, therapeutic hypothermia (TH) is currently the standard of care in developing countries for neonates presenting HIE. Therapeutic hypothermia consists of internal body temperature cooling to 33.5 °C for 72 h, beginning in the first 6 h after birth [17]. The hypothermic treatment timing seems critical since neonates that underwent TH up to 3 h after birth presented better outcomes [18].

According to a meta-analysis performed by Jacob et al. (2013) [6], TH reduces the absolute risk of mortality or neurodevelopmental disability in children with 18 to 24 months of age diagnosed with HIE at term by 15%. However, in newborns diagnosed with severe HIE, TH does not improve the major neurodevelopmental disabilities and neuromotor delays. Currently, TH is not established for preterm neonates presenting HIE due to the peculiarities of this injury when occurring in the preterm brain, and it was reported to be associated with a higher incidence of complications, such as hyperglycemia and coagulopathy, and higher mortality in preterm neonates [19,20]. Therefore, it is essential to uncover therapies that will improve the outcome of newborns diagnosed with HIE, including those classified as severe.

To address this concern, several adjuvant therapies to TH are currently being tested, such as erythropoietin, melatonin, topiramate, xenon, and stem cell therapy (SCT), among others [21]. In this systematic review, we will assess the potential of SCT for HIE, focusing on preclinical studies using stem cells isolated from the umbilical cord blood, umbilical cord tissue, placenta, and bone marrow.

## 3. The Potential of Stem Cell Therapy for HIE

Stem cells self-renew throughout life, and they can undergo differentiation to become specialized cells. Multipotent stem cells are undifferentiated cells that can self-renew and differentiate into distinct specialized cells of a certain lineage (e.g., osteogenic, adipogenic, chondrogenic, and hematopoietic, among others) [22]. There are several sources of multipotent stem cells in the human body, such as bone marrow, adipose tissue, and skin. Hematopoietic (HSCs) and mesenchymal stem/stromal cells (MSCs) are examples of multipotent stem cells found in adults [22].

The umbilical cord blood (UCB) and umbilical cord tissue (UCT) are excellent sources of stem cells, such as hematopoietic stem cells and mesenchymal stem/stromal cells, and other cells such as endothelial progenitor cells (EPCs) [23,24]. The UCB presents several advantages as a source of stem cells, especially in the case of hematopoietic transplantation: the collection is non-invasive and has no side effects for the newborn or the mother [25], uses tissue that is usually discarded, the cell isolation procedure is easy, and presents a reduced risk for graft-versus-host disease compared with stem cells isolated from other sources (e.g., bone marrow-derived stem cells) [26,27,28].

There is increasing evidence that stem cell therapy could have positive effects after a hypoxic-ischemic insult in the perinatal period. Several positive outcomes were identified in in vivo studies: improved functional outcome [29,30,31,32,33,34,35,36,37,38,39,40,41,42,43,44,45,46,47,48,49,50,51,52,53,54,55,56,57,58,59,60,61,62,63,64,65,66,67,68,69,70,71], increased angiogenesis [30,72], increased neurotrophic and growth factors levels [34,53,72,73,74], and cell proliferation [46,57,61]; reduction in the extension of brain damage [31,43,48,50,53,56,58,59,60,61,62,63,71,75,76], translated in decreased apoptosis [31,32,34,36,37,42,45,48,49,50,54,64,72,73,76,77,78]; decreased microglial activation and/or astrogliosis [32,34,35,36,37,40,42,43,45,46,48,49,50,54,57,62,69,73,75,77,78,79,80,81], and neuroinflammation [42,43,48,49,79,82] (Figure 1). Importantly, Cotten et al. (2014) and Tsuji et al. (2020) demonstrated, in two independent pilot feasibility and safety studies, that the collection, preparation, and intravenous infusion of a non-cryopreserved mononuclear fraction of cord blood cells is safe and feasible within the first postnatal days of newborns diagnosed with HIE [83,84]. Therefore, to assess the effects of SCT in animal models for HIE, the preclinical studies published in the last 10 years were summarized in Table 1, Table 2 and Table 3. The protocol for stem cell therapy varies considerably between studies, namely the type of stem cells used, and which tissue they were isolated from, the dosage of cells per injection, timepoint of treatment after insult, number of treatments applied, and route of administration; therefore, this section is divided into subsections based on the type of stem cell population used.

### 3.1. Umbilical Cord Blood Cells

The mononuclear fraction of the human UCB is a known source of different populations of stem and progenitor cells—hematopoietic stem cells, endothelial progenitor cells, and mesenchymal stem/stromal cells [23]. These cell populations can be identified through their surface marker profile and in vitro growth characteristics: HSCs are non-adherent, positive for CD34 and CD133; EPCs are adherent, positive for CD34, CD133, and CD90, and negative for CD13 and CD44; MSCs are adherent, positive for CD44, CD90, and CD13, and negative for CD34, CD45, and CD133 [23,88]

Several preclinical studies used the RV model for HIE [13] to assess treatment efficacy with UCB cells (Table 1). The experimental protocols applied in these studies were very heterogeneous, with the hypoxic period varying between 0.5 h to 3 h, the administration of UCB cells being conducted from 3 h post insult to 3 weeks post insult, and cell dosage per injection varying from 1.5 × 10^5^ cells to 10^8^ cells per injection. Different routes of UCB cell administration were also used (intravenous, intraventricular, intra-arterial, intranasal, and intraperitoneal). Nonetheless, most studies report a positive effect of UCB cell treatment in this model for HIE, namely a long-term recovery of the animals’ cognitive and motor functions [36,37,38,39,40,41,42,43,44,45,46,47]. Interestingly, two studies reported no improvement in the functional outcome after UCB cell administration [77,86]. However, the lack of a significant effect may be explained by the mild lesion induced, since the animals in this study were subjected to short hypoxic periods (0.5 h or 1 h).

The majority of the studies report a decrease in brain damage after administration of UCB cells, translated in decreased apoptosis [36,37,42,45,72,77,78] and neuronal loss [36,37,39,43,46,81,85,87]. Thus, the positive effects of UCB cells in the animals’ functional outcome can be linked to decreased brain damage. Nonetheless, some studies report an improvement of the functional outcome after UCB cell administration without observing a significant decrease in brain damage. Thus, other mechanisms are certainly influencing the long-term recovery observed after UCB cell treatment. Treatment with UCB cells decreased microglial activation [36,37,40,43,44,46,77,78,79] and astrogliosis [40,42,45,78], levels of pro-inflammatory cytokines [79] and oxidative stress [77], and increased the expression of growth factors in the central nervous system (CNC) [72], angiogenesis [50,72], neuronal stem cell (NSC) proliferation [85], and the number of mature neurons [42,72]. UCB cell treatment also seems to promote NSC differentiation into mature neuronal cells [81]. A study using an ovine model for HIE at term showed that autologous transplantation of UCB cells might contribute to the restoration of brain metabolic activity by reducing brain lactate levels [78]. Increased lactate levels were previously observed after the HI insult in the same model, correlating with neuronal injury [89].

Recently, Penny et al. (2020) showed that multiple doses of UCB cells administered at different timepoints, targeting the primary, secondary, and tertiary phases of the HI injury, were more effective in improving sensorimotor function and other parameters than a single dose at an earlier timepoint [36].

### 3.2. Mesenchymal Stem/Stromal Cells

Mesenchymal stem/stromal cells are multipotent stem cells present in distinct adult tissues, such as the bone marrow, and neonatal tissues, such as the placenta, umbilical cord tissue, and umbilical cord blood. All MSCs present potential for osteogenic, adipogenic, and chondrogenic differentiation [90], although cells from specific origins may have a stronger bias toward a specific lineage. Interestingly, MSCs isolated from neonatal tissues have improved proliferation, expansion, and transient engraftment capacity [91,92,93,94,95] and are less likely to induce an immune reaction [28]. Several criteria define MSCs, such as plastic adherence when maintained in culture; expression of CD105, CD73, and CD90; lack of expression of CD45, CD34, CD14 or CD11b, CD79a, or CD19, and HLA-DR surface molecules; and differentiation into osteoblasts, adipocytes, and chondroblasts in vitro [96].

Specific in vitro conditions can induce MSCs to differentiate into neuronal-like cells [97]. Some studies showed the differentiation of human MSCs into mature brain cells after a HI insult in the neonatal brain [33,51,55]. However, independent studies reported positive effects after MSC administration in HIE preclinical models, even in the absence of MSC differentiation into mature cell types [32,50,57,58]. An interesting research line would be to investigate the effect of MSC-derived secretome administration in HIE preclinical models. MSC-derived secretome showed to exert neuroprotective effects and/or stimulate paracrine endogenous repair mechanisms [98,99], and the intravenous administration of MSC-derived secretome improved motor recovery in a murine model of spinal cord injury and traumatic brain injury [100].

On the other hand, there is increasing evidence that in vitro MSC-preconditioning before transplantation/infusion enhances the efficacy of MSCs transplantation. There are several types of preconditioning being studied in the context of MSCs transplantation, such as hypoxia, the use of pharmacological agents, mechanotransduction stimuli, genetic engineering, among others [101,102]. For instance, mild-hypoxic preconditioning was shown to enhance the positive effects of MSC transplantation in an in vivo model of ischemic stroke by promoting neuronal repair, improving MSCs homing to the damaged tissues, and decreasing apoptosis [103]. Nonetheless, to the best of our knowledge, no study assessed the potential of preconditioned MSCs on in vivo models of HIE.

Another advantage of using MSCs for SCT is the in vitro expansion of the cells, which allows obtaining a higher number of cells for infusion.

#### 3.2.1. Umbilical Cord Tissue-Derived MSCs

Initially, the scientific community focused on the UCB as a valuable source of stem cells, and the umbilical cord tissue was considered medical waste. However, later in time, researchers realized that it was possible to isolate stem cells, namely MSCs, from various parts of the umbilical cord (e.g., Wharton’s jelly, cord lining, and the perivascular region) [104].

Umbilical cord tissue-derived MSCs (UCT-MSCs) differ from MSCs isolated from the bone marrow (BM-MSCs) by the higher proliferative potential [105], lower expression levels of vascular endothelial growth factor (VEGF) [106,107,108], and other proangiogenic factors [107], higher expression levels of chemokines and angiogenic growth factors [109,110], and higher secretion of neurotrophic factors [111]. UCT-MSCs also present a higher immunomodulatory activity than MSCs from other sources [105,112].

Similar to the treatment with UCB cells, UCT-MSCs treatment (Table 1) improved the functional outcome [29,30,31,32,33,34,35] and cellular morphology [30] and increased the number of mature neurons [29]. It also reported decreased brain damage [29,31,35,76] in some cases linked to the reduction of apoptosis [31,32,34,76], pro-inflammatory cytokine levels [76], astrogliosis [29,32,33,35], and microglial activation [29]. Additionally, Zhou et al. (2015) observed increased synaptic plasticity and angiogenesis with the infusion of UCT-MSCs after neonatal HI insult [30].

#### 3.2.2. Umbilical Cord Blood-Derived MSCs

MSCs are a poorly abundant stem cell population in the UCB; however, it is possible to successfully isolate MSCs from the UCB if the UCB is processed immediately after collection [113]. The UCB-derived MSCs (UCB-MSCs) are also multipotent and can differentiate into mesodermal, endodermal, and, importantly, in the context of HIE, ectodermal lineages [114,115]. Nonetheless, the ability to isolate MSCs from the UCB is not yet well established. Some authors showed the possibility of isolating MSCs from the UCB [116,117], while others disagree [118].

In the context of HIE, like other stem cell types, UCB-MSCs improve cognitive and motor function after HI insult in the perinatal and neonatal period [50,51], especially when combined with hypothermia (Table 1) [48,49]. Treatment with UCB-MSCs also decreased brain damage [49,50], translated into a reduction of the number of apoptotic cells [48,49,50], astrogliosis [48,50], and microglial activation levels [50].

#### 3.2.3. Placenta-Derived MSCs

The placental tissue represents an excellent source of progenitor/stem cells possessing abundant MSCs (PD-MSCs) readily available after birth, which are easily obtained [119,120]. In this review, two studies assessed the effects of PD-MSC administration in the murine model of HIE (Table 2) [52,53]. In one of them [53], PD-MSCs improved the animals’ functional outcome and physical appearance while decreasing the extent of brain damage and neuronal morphological changes induced by the HI insult. This treatment also reduced lipid peroxidation and free radical levels, which are hallmarks of HIE. Another study showed that intraventricular administration of PD-MSCs induced an immunomodulatory effect in the RV model for HIE by increasing the generation of regulatory T-cells, thus increasing anti-inflammatory cytokines and reducing pro-inflammatory cytokines in the brain and peripheral blood serum of the RV animal model. These observations were associated with decreased brain damage and improved functional outcomes [52].

#### 3.2.4. Bone Marrow-Derived MSCs

Several studies assessed the effects of bone marrow-derived mesenchymal stem/stromal cell administration in the murine model of HIE (Table 3).

Even though there were differences between the studies regarding the lesion severity and treatment protocol, the vast majority reported an improvement in motor and cognitive function after BM-MSCs administration [54,55,56,57,58,59,60,61,62,63,64,65,66,67,68,69,70]. The majority of the studies reported that BM-MSCs treatment decreased brain damage [54,56,58,59,60,61,62,63,67], prompting a decrease in apoptosis [54,64,73]. These positive effects were accompanied by the increase in cell proliferation [57,61], number of mature neurons count [56,67,80], anti-inflammatory cytokine levels [73], promotion of neuronal repair mechanisms [56]; and a decrease in microglial activation [54,57,62,67,80] and astrogliosis [62,69], as well as neuroinflammation [82].

### 3.3. Endothelial Progenitor Cells

Endothelial progenitor cells can also be isolated from the UCB. This cell type promotes vascular repair and tissue recovery after ischemia through the formation of new blood vessels. EPCs include a subtype of cells, the endothelial colony-forming cells (ECFCs), which have a high proliferating capacity and a specific vasculogenic activity [121].

**Table 3 ijms-22-03142-t003:** Studies focusing on the therapeutic potential of bone marrow-derived mesenchymal stem/stromal cells in animal models of hypoxic-ischemic encephalopathy.

Cell Type	Animal Model	Delivery Route (Source, Dose/Admin) Time of Treat.	Functional Outcome	Brain Damage/ Apoptosis	SC Engraftment	Other Outcomes/Observations	Ref.
BM-MSCs	Rat (RV; 3.5 h hypoxia)	IC (human; 10^6^ cells) 3 dpi	↑	=	Yes	MSCs transdifferentiated into astrocytes (astrocytic markers colocalized more with the transplanted MSCs than neuronal or oligodendrocyte markers).	[55]
	Mouse (RV; 0.75 h hypoxia)	ICV (murine; 10^5^ cells) 3 or 10 dpi	↑	↓	-	No difference between treatment at 10 and 3 dpi; increase in cell proliferation; no differentiation of MSCs into mature cell types; increase in the number of mature neurons, astrocytes, and oligodendrocytes; decrease in microglial activation.	[57]
	Mouse (RV; 0.75 h hypoxia)	IN (murine; 5 × 10^5^ cells) 10 dpi	↑	↓	Yes	No differentiation of MSCs into mature cell types.	[58]
	Mouse (RV; 0.75 h hypoxia)	ICV (murine; 10^5^ cells) 3 and/or 10 dpi	3 and 10 dpi ↑	3 and 10 dpi ↓	-	Injection at 3 dpi: Increase in mature neurons and oligodendrocytes count. Injection at 10 dpi: Increase in oligodendrocyte maturation, remyelination, and neuronal repair without new cell formation.	[56]
	Mouse (RV; 0.75 h hypoxia)	ICV (murine; 10^5^ cells) 3 and 10 dpi	↑	↓	-	Decrease in the HI-induced contralateral axonal rewiring and HI-induced changes in the white matter; increase in the axonal connectivity in the ipsilateral hemisphere.	[59]
	Rat (transient MCAO)	IN (murine; 10^6^ cells) 3 dpi	↑	↓	-	Increase in cell proliferation.	[61]
	Mouse (RV; 0.75 h hypoxia)	IN (murine; 0.25–1 × 10^6^ cells) 3/10/17 dpi	0.5 × 10^6^ ↑ 3 or 10 dpi ↑	0.5 × 10^6^ ↓ 3 or 10 dpi ↓ 17 dpi =	Yes	0.5 × 10^6^ cells: Lowest dose to produce alterations.	[60]
	Sheep Fetuses (0.4 h UCO)	IV (human; 3.5 × 10^6^ cells) 1 h after UCO	-	↓	Low	Decrease of the cerebral inflammatory response, T-cell invasion, and electrographic seizure activity; increase in persistent tolerance of T-cells and preOLs count.	[82]
	Mouse (RV; 0.75 h hypoxia)	IN (human; 1 or 2 × 10^6^ cells) 10 dpi	↑	1 × 10^6^ = 2 × 10^6^ ↓	Low	Decrease in astrogliosis and microglial activation.	[62]
	Mouse (RV; 0.75 h hypoxia)	IN (murine; 0.5 × 10^6^ cells) 10 dpi	↑	↓	-	No detection of adverse effects during the animal’s lifespan and no induction of tumors or other lesions in the brain or nasal turbinates.	[63]
	Rat (RV; 2.5 h hypoxia)	ICV (murine; 10^6^ cells) 2 dpi	↑	↓	-	Decrease in TLR2 expression levels.	[64]
	Rat (RV; 1.5 h hypoxia)	SC (murine; 0.75–1 × 10^6^ cells or 0.8–1.2 × 10^5^ cells) 7 dpi	↑	-	-	Increase in striatal medium spiny projection neurons—restored to uninjured levels with the higher dosage.	[65]
	Rat (RV; 2 h hypoxia)	ICV (murine; 2 × 10^5^ cells) 1 dpi	↑	-	Yes	Improvement in the neuronal pathological changes induced by the HI insult; increase in autophagy levels.	[66]
	Rat (RV; 1 h hypoxia)	IV (murine; 10^5^ cells) 1 dpi	↑	↓	-	Decrease in microglial activation.	[54]
	Rat (RV; 2 h hypoxia)	IV (murine; 10^6^ cells) 3 dpi	↑	↓	-	Increase in the number of neurons and synapses.	[67]
	Rat (RV; 1 h hypoxia)	IV (murine; 10^5^ cells) 4 hpi and 1 dpi	-	↓	-	Decrease in microglia activation (M1 phenotype); increase in anti-inflammatory cytokine and growth factor levels.	[73]
	Mouse (RV; 1 h hypoxia)	IN + TH (murine; 10^6^ cells) 3 dpi	MSCs/TH ↑ MSCs + TH ↑↑	MSCs + TH ↑	-	MSCs or TH: Decrease in growth factor expression levels. MSCs: Decrease in hypomyelination. MSCs + TH (compared to either therapy alone): Increase in the infiltration of endothelial cells and peripheral immune cells; increase in pro-inflammatory cytokines levels; decrease in growth factor expression levels (bellow control levels).	[68]
	Rat (RV; 2.5 h hypoxia)	IV (6 × 10^6^ cells) ?	-	-	Yes	Increase in HIF-1α and SDF-1α protein levels in the hippocampus.	[122]
	Mouse (RV; 0.75 h hypoxia)	IN (10^6^ cells) 10 dpi	-	↓	Yes	Increase in DCX^+^ cells in the SVZ, number of astrocytes at the lesion site, and the number of neurons; decrease in reactive astrocytes and microglial activation (M2 phenotype).	[80]
	Rat (bilateral ligation of cephalic arteries, 1.5 h hypoxia)	IV or ICV (3 × 10^6^ cells) 1 dpi	IV ↑ ICV ↑↑	IV ↓ ICV ↓↓	-	Decrease of astrogliosis.	[35]
	Rat (RV, 2.5 h hypoxia)	ICV (murine; 2 × 10^5^ cells) 5 dpi	↑	-	-	Enhanced long-term potentiation.	[70]
	Rat (RV, 2.5 h hypoxia)	ICV (murine; 2 × 10^5^ cells) 5 dpi	↑	-	-	Decrease in the number of proliferating astrocytes.	[69]

**Abbreviations:** ↑ increase or upregulation; ↓ decrease or downregulation; = no significant difference; - not evaluated; BM—bone marrow; DCX—doublecortin; hpi/dpi/wpi—hours post insult/days post insult/weeks post insult; IC—intracardiac; ICV—intraventricular; IN—intranasal; IV—intravenous; HI—hypoxic-ischemic; HIF—hypoxia-inducible factor; MCAO—middle cerebral artery occlusion; MSCs—mesenchymal stem/stromal cells; pre-OLs—oligodendrocyte progenitors; PX—postnatal day X; RV—Rice–Vannucci/Rice–Vannucci adaptation; SDF—stromal cell-derived factor; SVZ—subventricular zone; TH—therapeutic hypothermia; TLR—toll-like receptor; UCO—umbilical cord occlusion.

Regarding HIE, we found three different studies that assessed the effects of EPC/ECFC administration after a neonatal HI insult (Table 2). These studies reported that the administration of EPCs/ECFCs improved the animal’s functional outcome [71]; decreased brain damage [71], which can be linked to the decrease in apoptosis observed in the cortex ipsilateral to the lesion [42,43]; and increased the number of mature neuronal cells [42], cerebral capillary density, and cerebral blood flow [42]. Treatment with this cell population also hampered the inflammatory responses that occur after the neonatal HI insult [42,43].

## 4. Therapeutic Hypothermia and Stem Cell Therapy

Although TH is the current standard of care for HIE in term neonates in developing countries, it is not entirely effective in preventing mortality or neurodevelopmental disabilities in HIE patients, especially those diagnosed with severe HIE [6]. Therefore, it is crucial to find safe and effective therapies that will enhance TH’s neuroprotective effects and improve the outcome of these patients.

To our knowledge, few preclinical studies assessed the potential of combining TH with SCT [48,49,68] to treat severe HIE. These studies present some contradictory results. Two studies revealed that hypothermia alone did not improve the animals’ functional outcome following severe HIE [48,49]; however, hypothermia and MSCs infusion 2 days after insult had a positive effect, improving the animal’s functional outcome while decreasing brain damage, cytokine levels, microgliosis, and astrogliosis [48,49]. Interestingly, both studies reported that combined therapy was more effective than MSC administration alone [48,49].

In contrast, a study performed by Herz et al. (2018) showed that animals treated either with TH or MSCs had a better outcome than animals treated with the combined therapy of TH followed by MSC administration 3 dpi [68]. This study revealed that, after HI insult in the neonatal period, only the MSC treatment improved cognitive function and decreased white matter injury, and MSC or TH treatment improved motor function. However, the combined therapy, TH followed later by MSC administration, reversed the protective effects observed with each therapy alone, resulting in increased, long-lasting functional deficits, brain damage, endothelial cells infiltration, peripheral immune cell infiltration, and pro-inflammatory cytokine levels, as well as decreased levels of growth factor expression. One potential mechanism pointed out by the authors is an alteration of the cerebral microenvironment after TH, resulting in a modification of the MSCs phenotype after their administration. This alteration may induce pro-inflammatory cytokine expression and block the expression of growth factors, thus interfering with the rescuing of the injured brain.

## 5. Mechanisms of Action of SCT after HI Insult in the Developing Brain

This section summarizes and further discusses the previously identified mechanisms of action of SCT in the preclinical studies included in this systematic review (Figure 1). Several mechanisms of action might be mediating the positive effects observed after SCT in animals subjected to a HI insult in the developing brain. These positive effects are most likely not due to a particular mechanism but due to a synergistic or cumulative effect. Stem cells might exert their action by protecting the brain from injury and enhancing endogenous repair mechanisms.

### 5.1. Stem Cell Engraftment and Differentiation into Mature Neuronal Cells

Stem cell engraftment at the lesion site was reported by several studies [31,32,33,38,39,40,50,51,55,58,60,71,75,76,80,122], which in most of the cases was transient, while others reported low or no engraftment [37,46,47,53,62,82]. Additionally, only three studies reported stem cell differentiation into mature neuronal cells [50,51,55]. Thus, considering the long-term effects observed after stem cell administration, the engraftment and differentiation of stem cells appear to have little influence on the observed positive effects of SCT in HIE preclinical models.

### 5.2. Secreted Factors and Paracrine Effects

UCB cells and MSCs secrete a wide range of factors that contribute to damaged tissue regeneration, such as angiogenic factors, chemokines, and neurotrophic factors [123,124,125]. Moreover, several studies reported increased levels of these factors after SCT in HIE preclinical models [34,53,72,73,74]. Thus, it is likely that the beneficial effect of SCT might also be related to the presence of these paracrine factors, or secretome, contributing to neuronal repair, increasing angiogenesis, hampering the inflammatory response, and promoting an anti-apoptotic effect, among others.

### 5.3. Neurogenesis

An increased number of mature neurons was reported after SCT [29,42,57,58,67,72,80,81,87] along with cell proliferation [46,57,61]. Additionally, two studies showed that systemic administration of stem cells stimulated NSC proliferation [85] and differentiation [81]. Therefore, SCT might be stimulating endogenous neurogenesis, contributing to the neuronal repair after HI insult in the developing brain.

### 5.4. Apoptosis

SCT has been described to have an anti-apoptotic effect in HIE preclinical models, which contributes to a decrease in brain damage, explaining the positive functional outcomes observed [31,32,34,36,37,42,45,48,49,50,54,64,72,73,76,77,78].

### 5.5. Astrogliosis and Activated Microglia

Decreased astrogliosis [32,35,40,42,45,48,50,62,69,75,78,80,81] and/or microglial activation [34,36,37,40,43,46,48,49,50,54,57,62,73,75,77,78,79,80] have been reported after stem cell administration. Considering that the HI insult in the developing brain triggers astrogliosis and microglial activation, contributing to increased brain damage, SCT might exert its protective effects by decreasing the levels of these two processes.

### 5.6. Inflammation

SCT was associated with a significant decrease in pro-inflammatory cytokines [42,52,73,79,82]. This effect was further enhanced by combining SCT with TH [48]. Thus, it appears that some positive effects of SCT in HIE preclinical models might be due to the downregulation of the pro-inflammatory cytokines that were augmented after the HI insult in the developing brain.

### 5.7. Angiogenesis

Increased angiogenesis and increased levels of pro-angiogenic factors, such as VEGF and interleukin 8, were reported by two studies after SCT [30,72].

### 5.8. Oxidative Stress

An important hallmark of HIE is oxidative stress, and stem cell administration in a preclinical model was reported to decrease the number of cells positive for oxidative stress markers [77]. Nonetheless, this observation was not accompanied by long-term functional and morphological improvements.

## 6. Analysis and Discussion

From the 58 studies included in this systematic review (Appendix A), we identified 83 different protocols for stem cell therapy in animal models for HIE (Appendix A). The most common animal model employed was the Rice–Vannucci model, with 88% of the studies choosing this model to induce HI insult in the developing brain (*n* = 51; Figure 2A). Rodents were predominantly used in the included studies (Figure 2B), the rat being the most used species (*n* = 42), followed by mouse (*n* = 13).

Regarding the sex of the animals (Figure 2C), 43% of the studies did not report this information (*n* = 25), 36% used both females and males (*n* = 21), and 21% used only males (*n* = 12). None of the included studies investigated the effect of SCT in females alone, nor did a comparative analysis.

As expected, most of the studies (90%) assessed the effect of SCT on histological parameters (*n* = 52; Figure 3). However, fewer studies assessed the animal’s functional outcome following SCT: 64% evaluated the animal’s sensorimotor function (*n* = 37), and only 29% evaluated the animal’s cognitive function (*n* = 17) (Figure 3). The evaluation of the functional outcome after SCT is of great importance for the clinical setting since one of the primary goals is to tackle the cognitive and motor impairments caused by HIE. Additionally, different behavioral tests were used at different time points: 17 different tests were used to evaluate the sensorimotor function, the cylinder rearing test (*n* = 19) and the rotarod (*n* = 11) being the most used ones; and three different tests were used to evaluate cognitive function: the Morris water maze (*n* = 10), the novel object recognition test (*n* = 5), and the object in place task (*n* = 2) (Appendix A). These disparities make a comparison between studies a challenge. Considering only the studies that evaluated sensorimotor function, cognitive function, and/or histological parameters, about 80% of the protocols included in these studies prompted a significant improvement in at least one of these parameters. However, due to the lack of reporting the complete data values for each experimental group (e.g., mean and standard error of the mean), we could not identify which protocol(s) represented the best strategy for neurological and neurobehavioral recovery after a HI insult in the neonatal period.

Focusing on the studies using the RV model in rats (67%, *n* = 39), the duration of the hypoxic insult—that is, the exposure to 8% O_2_—ranged between 1 and 4 h (Figure 4), with 50% of the studies applying a hypoxic insult with a duration between 1.5 and 2.5 h (*n* = 29). Considering these studies, which most likely had a similar lesion extent, we selected the ones conducted in postnatal day-7 rats since they represent 45% of the studies (*n* = 26) (Appendix A).

From the 26 studies that established the HIE model using the RV protocol at P7, 19 used cells isolated from neonatal tissues. These 19 studies evaluated 25 protocols (Appendix A), which included the usage of distinct cell types isolated from the umbilical cord (blood or tissue) or the placenta: UCB cells (14 protocols), UCB-MSCs (3 protocols), UCT-MSCs (6 protocols), and PD-MSCs (2 protocols) (Figure 5).

Regarding UCB cell transplantation, the most used routes of administration were intraventricular injection (4 protocols), intravenous (4 protocols), and intraperitoneal administration (4 protocols) (Appendix A). The minimal dose of UCB cells eliciting a significant motor function improvement and a decrease in brain damage was 10^6^ cells, administered intravenously seven days post insult [45]. In contrast, a different study, which compared the effect of different doses of UCB cells administered intravenously (10^6^, 10^7^, and 10^8^ cells) found no improvement of the animal’s cognitive function and brain damage with one administration of 10^6^ cells [39]. Nonetheless, in this study, UCB cells were administered at a different time point, one day post insult, and the animal’s motor function was not evaluated [39].

For MSCs derived from neonatal tissues, the predominant route of administration was intraventricular injection (8 protocols), followed by intravenous (2 protocols) and intraperitoneal administration (1 protocol) (Appendix A). Although the ICV was the predominant administration route, it has little translatability to the clinical practice due to its invasiveness and the risks associated with this procedure. The efficacy of intravenous administration of MSCs was evaluated in a study that compared the administration at two time points and concluded that a single intravenous administration of 5 × 10^5^ cells one day post insult was significantly more effective in improving the animal’s motor and cognitive function than the same dose administered three days post insult [33].

Nine included studies evaluated the efficacy of intranasal administration of UCB cells [36], UCT-MSCs [29], and BM-MSCs [58,60,61,62,63,68,80]. All these studies reported histological and functional rescue of animals subjected to a HI insult in the developing brain using this route of administration. Importantly, Donega et al. (2015) showed that intranasal administration of MSCs was long-term effective and safe in mice subjected to neonatal HI [63]. Moreover, intranasal administration appears to have efficacy in low doses of MSCs. A study showed positive histological and functional outcomes even after one administration of 2 × 10^5^ UCT-MSCs [29]. The therapeutic window for intranasal administration of MSCs appears to be up to 10 days, even with lower doses [58,63]. On the other hand, Penny et al. (2020) found no significant difference between the effectiveness of intraperitoneal or intranasal administration of UCB cells in the RV model for neonatal HIE [36].

Overall, SCT appears to improve the HIE animal model’s sensorimotor and cognitive function and decreases brain damage. Nonetheless, there is a high variability since it used 7 types and sources of stem cells, 18 dosages, 14 time points, and 8 administration routes, suggesting the need for further studies to elucidate the ideal procedure.

## 7. Conclusions and Future Perspectives

We systematically reviewed studies reported in the last 10 years assessing the potential of using cells isolated from the UCB, UCT, placenta, and BM in preclinical animal models for HIE. The positive effects reported included improved functional outcome, both cognitive and motor function, decreased brain damage, translated by a decrease in apoptotic cells and prevention of neuronal loss, microglial activation, astrogliosis, inflammation, and increased angiogenesis and cell proliferation, among others. Thus, stem cell therapy appears to have great therapeutic potential and could become a new therapy for HIE. Nonetheless, there is a high variability regarding the dose of stem cells applied, route, and administration timing. Therefore, it would be critical to perform studies assessing different amounts of stem cells, considering the clinical setting, and determining the optimal time for stem cell administration (e.g., if during the secondary or tertiary phase of the injury) to increase the chance of successful translating stem cell therapy into the clinical practice.

A new possible therapeutic combination would be adding SCT to the current standard of care for HIE, TH, thus improving the effectiveness of TH in treating infants diagnosed with HIE, especially those diagnosed with severe HIE. However, the lack of studies addressing the effect of combining TH with SCT in HIE and the existing heterogeneity in the few studies that were performed until today stresses the importance of exploring this research line.

In conclusion, there is increasing evidence in the literature that SCT could, in combination with TH, be the next standard of care for HIE patients, addressing the lack of effectiveness of therapeutic hypothermia. Infusion of human umbilical cord blood cells was already demonstrated to be safe and feasible in newborns diagnosed with HIE. However, it is still necessary to optimize the protocol for SCT, namely determining the optimal dose, route of administration, and timing, as well as assessing which stem cell types provide the maximal neuroprotection. This is where translational research and animal models become extremely useful, allowing them to explore multiple therapeutic interventions and unravel which ones have the potential to be applied in the clinic.

## 8. Methods

### 8.1. Literature Search

The literature search was performed using MEDLINE’s database, PubMed, and Web of Science on 27 October 2020. Search terms included “hypoxic-ischemic encephalopathy”, “stem cells”, “umbilical cord cells”, “umbilical cord blood cells”, and “mesenchymal stem cells”, and other synonyms of these words (Figure 5). Studies in duplicate were manually removed from the search results, as well as studies in languages other than English. Studies published before 2010 were excluded, and only full-text articles were included. Relevant review articles were also manually searched to maximize the inclusion of relevant studies. Two authors screened the abstracts and the full text of the studies independently. Disagreements were resolved by discussion.

### 8.2. Inclusion and Exclusion Criteria

The literature search and the screening processes used are summarized in Figure 6. Only preclinical studies were included in this systematic review, excluding reviews, chapters, clinical studies or case reports, and in vitro studies. Studies were included if in vivo preclinical models of HIE were used, namely occlusion of the carotid artery, middle cerebral artery, uterine artery, or umbilical cord. Studies reporting the effect of stem cells isolated from the umbilical cord, umbilical cord blood, placenta, and bone marrow were included. Other stem cells (e.g., neuronal stem cells, multipotent adult progenitor cells) or stem cells isolated from other adult tissues were excluded. Studies that did not evaluate stem cell efficacy or use modified stem cells, except for tracing and locating the distribution of the cells, were excluded. Studies that assessed the effect of other therapies in combination with stem cell administration were excluded, except if the applied therapy was therapeutic hypothermia since it is the current standard of care for HIE.

### 8.3. Data Extraction

From the included studies, the following data were collected and analyzed: general study design, animal characteristics (animal model, species, sex, age, and weight), the protocol for SCT (source of stem cells, stem cell processing before administration, dose, administration route, timepoint of administration, the number of administrations, and amount of cells/administration), histological techniques and neurobehavioral tests used, and respective outcomes. Appendix A summarizes the data extraction for all studies included in this systematic review, and Appendix A lists all the protocols for stem cell therapy included in the 58 studies.

## Figures and Tables

**Figure 1 ijms-22-03142-f001:**
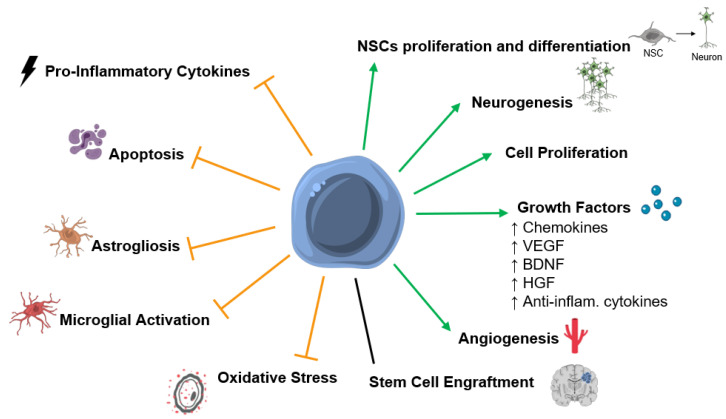
Mechanisms of action of stem cells identified by the preclinical studies included in this systematic review. Several mechanisms of action that might be mediating the positive functional outcomes observed after SCT in preclinical models of neonatal hypoxic-ischemic encephalopathy (HIE). Stem cell therapy (SCT) was associated with the promotion or upregulation (green arrows) of neuronal stem cells (NSCs) proliferation and differentiation, neurogenesis, cell proliferation, growth factors levels/secretion, angiogenesis, and inhibition or downregulation (orange truncated arrows) of pro-inflammatory cytokines, apoptosis, astrogliosis, microglial activation, and oxidative stress. Also, some studies report stem cell engrafment after SCT, while other report low or no engrafment. Abbreviations: Anti-inflam—anti-inflammatory; BDNF—brain-derived neurotrophic factor; HGF—hepatocyte growth factor; VEGF—vascular endothelial growth factor.

**Figure 2 ijms-22-03142-f002:**
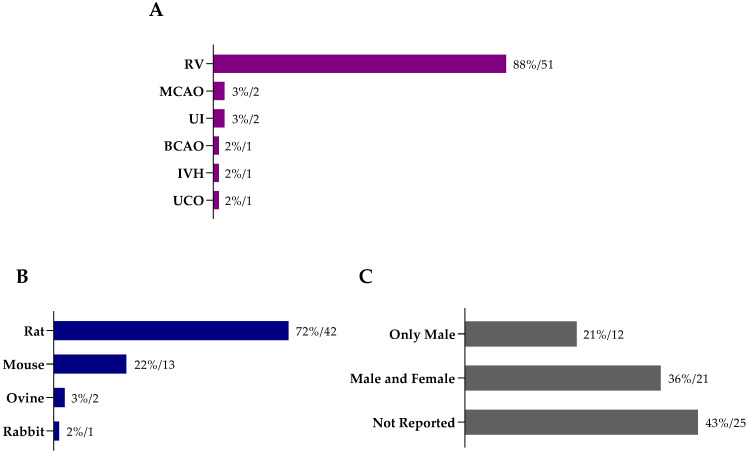
(**A**) Animal models used to induce hypoxic-ischemic encephalopathy in the studies included in this systematic review, as well as the employed (**B**) species and (**C**) animals’ sex across the different studies (percentage/number of reports). Abbreviations: BCAO—bilateral carotid artery occlusion; IVH—intraventricular hemorrhage; RV—Rice–Vannucci animal model (or adaptation); MCAO—middle cerebral artery occlusion; UCO—umbilical cord occlusion; UI—uterine ischemia.

**Figure 3 ijms-22-03142-f003:**
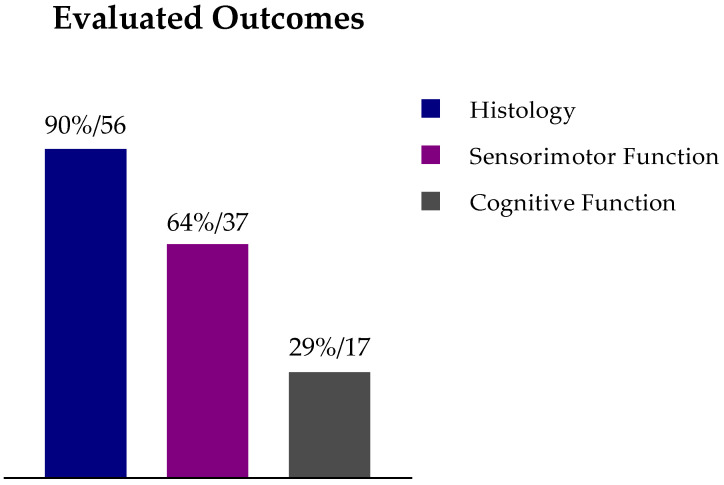
Studies that included histological, cognitive function, and/or sensorimotor function evaluation after stem cell therapy in animal models for hypoxic-ischemic encephalopathy (percentage/number of reports).

**Figure 4 ijms-22-03142-f004:**
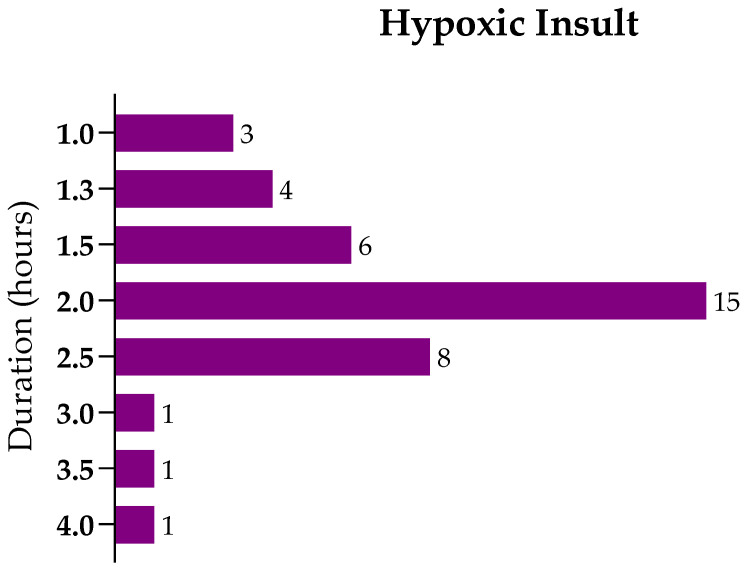
Distribution of the studies using the Rice–Vannucci protocol to induce hypoxic-ischemic brain lesion in neonatal rats regarding the hypoxic insult’s duration in hours.

**Figure 5 ijms-22-03142-f005:**
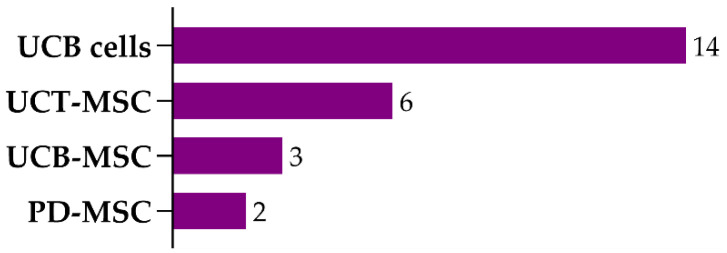
Number of protocols using different cell types isolated from neonatal tissues used in postnatal day-7 rats subjected to unilateral carotid artery occlusion, followed by 1.5–2.5 h of hypoxia. Abbreviations: UCB cells—umbilical cord blood cells; UCT-MSCs—umbilical cord tissue mesenchymal stem/stromal cells; UCB-MSCs—umbilical cord blood mesenchymal stem/stromal cells; PD-MSCs—placenta-derived mesenchymal stem/stromal cells.

**Figure 6 ijms-22-03142-f006:**
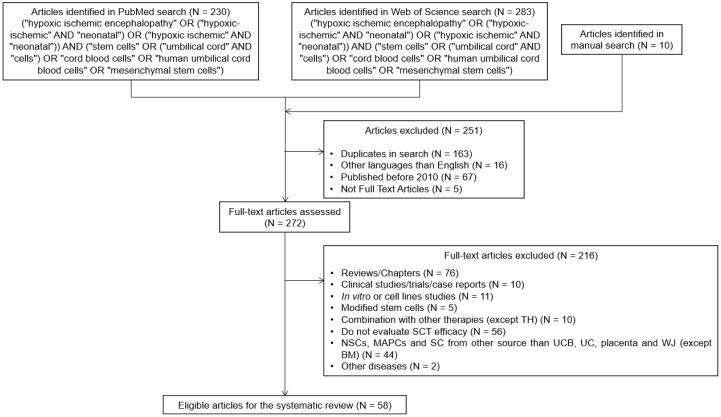
Flow diagram representing the literature search and screening processes applied to select the preclinical studies included in this systematic review. Abbreviations: BM—bone-marrow; MAPCs—multipotent adult progenitor cells; NSCs—neuronal stem cells; SC—stem cells; UC—umbilical cord; UCB—umbilical cord blood; WJ—wharton’s jelly.

**Table 1 ijms-22-03142-t001:** Studies focusing on the therapeutic potential of umbilical cord blood cells and umbilical cord blood and umbilical cord tissue stem/stromal cells from human origin (except when mentioned otherwise) in animal models for hypoxic-ischemic encephalopathy.

Cell Type	Animal Model	Delivery Route (Dose/Admin) Time of Treat.	Functional Outcome	Brain Damage/ Apoptosis	SC Engraftment	Other Outcomes/Observations	Ref.
UCB cells	Rat (RV; 1.5 h hypoxia)	IP (2 × 10^6^ cells) 3 hpi	↑	↓	Low	Decrease in microglial activation and number of macrophages.	[37]
	Rat (RV; 1.3 h hypoxia)	IP (10^7^ cells) 1 dpi	↑	=	Yes	Restoration of cortical plasticity.	[38]
	Rat (RV; 1.3 h hypoxia)	IP (10^7^ cells) 1 dpi	-	↓	-	Increase in mature neuron count, angiogenesis levels, occludin levels, and growth factor expression.	[72]
	Rat (RV; 2 h hypoxia)	IV (10^6^/10^7^/10^8^ cells) 1 dpi	10^8^ ↑	10^7^ or 10^8^ ↓	Yes		[39]
	Rat (RV; 1.3 h hypoxia)	IP or IT (10^7^ cells) 1 dpi	↑	-	Yes	Both administration routes led to similar outcomes; decrease in microglial activation, astrogliosis, and invading macrophages.	[40]
	Rat (RV; 2 h hypoxia)	ICV (3 × 10^6^ cells) 1 dpi	-	↓	-	Increase in NSC proliferation.	[85]
	Rat (RV; 1.3 h hypoxia)	IP (10^7^ cells) 1 dpi	-	-	-	Decrease in pro-inflammatory cytokines serum levels, microglial activation, and macrophage infiltration.	[79]
	Rat (RV; 2 h hypoxia)	IA (10^6^/10^7^ cells) 1 dpi	**10^7^** ↑	=	-		[41]
	Rat (RV; 1 h hypoxia)	IP (10^7^ cells) 6 hpi	=	Apoptosis ↓ Brain damage =	-	Decrease in oxidative stress and microglial activation.	[77]
	Rat (RV; 1.5 h hypoxia)	IP (10^7^ cells) 2 dpi	↑	↓	-	Decrease in neuroinflammation; increase in the number of mature neurons; delayed glial scar formation (12 wpi); increase in cerebral capillary density and cerebral blood flow.	[42]
	Rat (RV; 3 h hypoxia)	IP (10^6^ cells) 1 dpi	↑	Apoptosis = Brain damage ↓	-	Decrease in infiltrating CD4^+^ T-cells, number of T-cells with pro-inflammatory phenotype, and microglial activation; no alteration of growth factor expression levels.	[43]
	Rat (RV; 3 h hypoxia)	IP (10^6^ cells) 1 dpi	↑	=	-	Increase in microglial activation in the somatosensory cortex.	[44]
	Rat (RV; 1.5 h hypoxia)	IP/IN (10^6^ cells) 1 dpi or 1, 3, and 10 dpi	1 + 3 + 10 dpi ↑	1 + 3 + 10 dpi ↓	-	Multiple doses: Decrease in microglial activation.	[36]
	Rat (RV; 2.5 h hypoxia)	IV (1.5 × 10^4^ CD34^+^ cells or 10^6^ MNCs) 7 dpi	↑	↓	-	Decrease in cerebral atrophy and astrogliosis; increase in DCX and lectin expression.	[45]
	Rat (RV; 2 h hypoxia)	IP (murine; 2 × 10^6^ cells) 3 dpi	↑	↓	Low	Increase in proliferating cells in the hippocampus; decrease in microglial activation and macrophage infiltration.	[46]
	Mouse (RV; 0.5 h hypoxia)	IV (10^5^ CD34^+^ cells) 2 dpi	=	=	-	Increase in CBF in the ischemic penumbra.	[86]
	Rabbit (0.67 h UI)	IV (5 × 10^6^ cells or 2.5 × 10^6^ cells) 4 h after birth	2.5 × 10^6^ ↑ 5 × 10^6^ ↑↑	-	No		[47]
	Rat (RV; 2 h hypoxia)	ICV (3 × 10^6^ cells) 1 dpi	-	-	-	Increase in the number of neurons and NSC differentiation into neuronal cells; decrease in the number of glial cells and glial differentiation.	[81]
	Mouse (RV; 2 h hypoxia)	IV (5 × 10^6^ cells) 3 wpi	-	-	-	Administration of UCB cells induced a shift in chemokine expression profile after HI insult; increase in chemokine levels to the damaged brain tissue.	[74]
	Rat (RV, 2 h hypoxia)	ICV (3 × 10^6^ cells) 1 dpi	-	-	-	Increase of neuronal cell count; decrease in TLR4 protein levels and NF-kβ protein levels; decrease in IL-1β level in the ipsilateral cortex.	[87]
	Lamb (UCO)	IA (10^8^ cells) 12 hpi	-	↓	-	Decrease in astrogliosis, microglial activation, and macrophage infiltration; restoration of normal brain metabolism.	[78]
UCT-MSCs	Rat (PWMD; 4 h hypoxia)	IP (10^6^ cells) 1, 2, and 3 dpi	↑	↓	Yes	Decrease in astrogliosis and microglial activation.	[75]
	Rat (RV; 2 h hypoxia)	ICV (2 × 10^5^ cells) 5 dpi	↑	↓	-	Increase in hippocampal synaptic plasticity, IL-8 protein levels, and angiogenesis in the hippocampus.	[30]
	Rat (RV; 2 h hypoxia)	ICV (3 × 10^5^ cells) 3 dpi	-	↓	Yes	Decrease in TNF-α and IL-1β expression levels in the damaged nerve cells.	[76]
	Rat (RV; 2 h hypoxia)	ICV (5 × 10^3^ cells) 1 hpi	↑	↓	Yes		[31]
	Rat (RV; 2 h hypoxia)	ICV (10^6^ cells) 1 dpi	↑	↓	Yes	MSCs did not differentiate into neuronal or glial cells; decrease in astrogliosis.	[32]
	Rat (RV; 2 h hypoxia)	IV/IP (IV: 5 × 10^5^ cells IP: 5 × 10^6^ cells) 1 or 3 dpi	↑ (Best 1 dpi)	-	Yes	IV route: More MSCs detected in the frontal cortex; greatest decrease in astrogliosis.	[33]
	Rat (RV; 1.5 h hypoxia)	IN (2 × 10^5^ cells) 1 dpi	↑	↓	-	Increase in the number of neurons; decrease in astrogliosis and microglial activation; no alteration of BDNF expression levels.	[29]
	Mouse (IVH model)	IV (10^5^ cells) 2 dpi	↑	↓	-	Decrease of reactive gliosis, hypomyelination, and periventricular cell death; increase in BDNF and HGF expression levels in the serum, CBF, and brain tissue.	[34]
UCB-MSCs	Rat (RV; 2.5 h hypoxia)	ICV (10^5^ cells) 3 dpi	↑	-	Yes	Some MSCs differentiated into astrocyte-like cells.	[51]
	Rat (Permanent MCAO)	ICV (10^5^ cells) 6 hpi	↑	↓	Yes	Decrease in astrogliosis and microglial activation; decrease in mortality; few MSCs differentiated into neuronal or glial cells.	[50]
	Rat (RV; 2 h hypoxia)	ICV + TH (10^5^ cells) 6 hpi	MSCs + TH ↑	MSCs ↓ MSCs + TH ↓↓	-	MSCs + TH (compared with each treatment alone): Decrease in microglial activation and inflammatory cytokines levels.	[49]
	Rat (RV; 2 h hypoxia)	ICV + TH (10^5^ cells) 2 dpi	MSCs + TH ↑	MSC ↓ MSCs + TH ↓↓	-	MSCs: Decrease of astrogliosis. MSCs + TH (compared with each treatment alone): Decrease in inflammatory cytokines levels; greatest decrease in astrogliosis and microglial activation.	[48]

**Abbreviations:** ↑ increase or upregulation; ↓ decrease or downregulation; = no significant difference; - not evaluated; BDNF—brain-derived neurotrophic factor; CBF—cerebral blood flow; DCX—doublecortin; hpi/dpi/wpi—hours post insult/days post insult/weeks post insult; HSCs—hematopoietic stem cells; IA—intra-arterial; IC—intracardiac; ICV—intraventricular; IL—interleukin; IN—intranasal; IP—intraperitoneal; IT—intrathecal; IV—intravenous; IVH—intraventricular hemorrhage; HGF—hepatocyte growth factor; HI—hypoxic-ischemic; MCAO—middle cerebral artery occlusion; MNC—UCB mononuclear fraction; MSCs—mesenchymal stem/stromal cells; NF-kβ—nuclear factor kappa of activated B cells; NSCs—neuronal stem cells; PWMD—periventricular white matter damage; PX—postnatal day X; RV—Rice–Vannucci/Rice–Vannucci adaptation; TH—therapeutic hypothermia; UCB cells—human umbilical cord blood cells (mononuclear fraction); UCT—umbilical cord tissue; UCO—umbilical cord occlusion; UI—uterine ischemia; TLR—toll-like receptor; TNF—tumor necrosis factor.

**Table 2 ijms-22-03142-t002:** Studies focusing on the therapeutic potential of placenta-derived mesenchymal stem/stromal cells and endothelial progenitor cells/endothelial colony-forming cells from human origin in animal models of hypoxic-ischemic encephalopathy.

Cell Type	Animal Model	Delivery Route (Dose/Admin) Time of Treat.	Functional Outcome	Brain Damage/ Apoptosis	SC Engraftment	Other Outcomes/Observations	Ref.
PD-MSCs	Rat (RV; 2.5 h hypoxia)	ICV (10^6^ cells) 2 dpi	↑	↓	No	Improvement of neuronal morphological changes induced by the HI insult; further increase in ROS levels; decrease in lipid peroxidation and free radical generation.	[53]
	Rat (RV, 2.5 h hypoxia)	ICV (5 × 10^4^ cells) 2 dpi	↑	↓	-	Decrease in pro-inflammatory cytokines expression levels in the ipsilateral hemisphere and in serum; increase in anti-inflammatory cytokine levels in the serum and the proportion of Treg cells;	[52]
EPCs/ECFCs	Mouse (RV; 0.5 h hypoxia)	IP (10^5^ cells) 1 dpi	↑	↓	Yes	Decrease in the size of the cystic lesion.	[71]
	Rat (RV; 1.5 h hypoxia)	IP (5 × 10^5^ cells) 2 dpi	↑	↓	-	Increase in the number of mature neuronal cells in the ipsilateral hemisphere; decrease in late astrogliosis; increase in cerebral capillary density and cerebral blood flow.	[42]
	Rat (RV; 3 h hypoxia)	IP (2 × 10^5^ cells) 1 dpi	↑	↓	-	Decrease in infiltrating CD4^+^-T-cells (to uninjured levels), the number of T-cells with pro-inflammatory phenotype, and microglial activation.	[43]

**Abbreviations:** ↑ increase or upregulation; ↓ decrease or downregulation; = no significant difference; - not evaluated; dpi—days post insult; EPCs/ECFCs—endothelial progenitor cells/endothelial colony-forming cells; ICV—intraventricular; IP—intraperitoneal; HI—hypoxic-ischemic; MSCs—mesenchymal stem/stromal cells; PD—placenta-derived; ROS—reactive oxygen species; RV—Rice–Vannucci/Rice–Vannucci adaptation; Treg—regulatory T-cells.

## Data Availability

No new data were created or analyzed in this study. Data sharing is not applicable to this article.

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
