# Peer review of "Stem Cell Therapy for Neonatal Hypoxic-Ischemic Encephalopathy: A Systematic Review of Preclinical Studies"

_ijms, 2021, doi:10.3390/ijms22063142_

Round 1
Reviewer 1 Report
The manuscript presented by the Authors is scientifically sound and well presented.
I only have minor revisions suggested to improve the overall quality of the manuscript.
- In the abstract, please highlight the systematic review was in preclinical studies
- Line: 54 Please provide reference for this incidence. The majority of developed countries report incidences no higher than 6/1,000.
- Line 84: When discussing the predisposition to injury the Authors should also mention the reduced antioxidant 'buffering' capacity of the developing brain which results in significant oxidative injury.
See: doi.org/10.1089/ars.2010.3581. - Line 99: Brain injury is also highly dependent on vessel occlusion as well.
See: doi.org/10.1016/j.jneumeth.2017.06.016.
Where occlusion of both the CCA and ECA results in a reliable and large infarct. Whereas the CCA only results in a variable infarct. - Line: 108 TH has to be initiated within 6 hours of HIE onset, not 72h.
- Line 124: See doi.org/10.1016/j.jpeds.2016.11.019.
Rao et al discuss limitations of TH in preterm infants as well.
Author Response
The manuscript presented by the Authors is scientifically sound and well presented.
I only have minor revisions suggested to improve the overall quality of the manuscript.
1. In the abstract, please highlight the systematic review was in preclinical studies.
Line 28, we changed the sentence to "With this systematic review, we gathered information included in 58 preclinical studies from the last decade…"
2. Line: 54 Please provide reference for this incidence. The majority of developed countries report incidences no higher than 6/1,000.
The incidence that we mention of 8.5/1000 reports a global incidence for neonatal encephalopathy (NE) related to intrapartum hypoxic events in 2010 (doi:10.1038/pr.2013.206). However, after further research, we found that HIE and NE incidence is highly variable across studies; thus, we altered the phrase and cited a review that discusses this matter (10.1016/j.earlhumdev.2010.05.010). We also added the incidence of neonatal HIE in developed countries, which is approximately 0.5 to 1 case per 1000 births, referencing the Cochrane review (Jacobs et al. 2013).
Line 55-60: "The estimated incidence of HIE is variable across studies [5]. In developed countries, neonatal HIE incidence is approximately 0.5 to 1 case per 1000 live-births [6], however, the global estimate is highly influenced by the higher incidence found in developing countries [7].
3. Line 84: When discussing the predisposition to injury the Authors should also mention the reduced antioxidant 'buffering' capacity of the developing brain which results in significant oxidative injury.
See: doi.org/10.1089/ars.2010.3581.
We agreed that this information is important to mention and added to the sentence, citing the paper brought up to us by the reviewer.
Line 80: "Due to poor antioxidant defenses and higher fatty acid concentrations, the developing brain is more susceptible to oxidative stress, therefore being highly vulnerable to hypoxic-ischemic insults [10].”
4. Line 99: Brain injury is also highly dependent on vessel occlusion as well.
See: doi.org/10.1016/j.jneumeth.2017.06.016.
Where occlusion of both the CCA and ECA results in a reliable and large infarct. Whereas the CCA only results in a variable infarct.
We appreciated that the reviewer brought this paper to our attention and added this information to the sentence.
Line 102: "The degree of brain damage severity is highly dependent on both of the duration of the hypoxic period [16], and the arteries occluded, since the ligation of both the common and external carotid arteries, instead of only CCA ligation, results in a larger and more consistent brain lesion [17]."
5. Line: 108 TH has to be initiated within 6 hours of HIE onset, not 72h.
We agree with the reviewer, the information was incorrect. We deleted the sentence since this timepoint (6 hours) was correctly mentioned in the sentence below.
Line 114-115: sentence deleted
Line 124: See doi.org/10.1016/j.jpeds.2016.11.019.
Rao et al discuss limitations of TH in preterm infants as well
Since TH is not established for preterm neonates presenting HIE and this type of injury involves distinct mechanisms, when compared to term HIE, we decided to not explore this topic. Nonetheless, we added a short sentence and cited the suggested study and a more recent review (doi: 10.3390/ijms22041671).
Line 128-131: "Currently TH is not established for preterm neonates presenting HIE, due to the peculiarities of this injury when occurring in the preterm brain, and it was reported to be associated with a higher incidence of complications, such as hyperglycemia and coagulopathy, and higher mortality in preterm neonates [21, 22]."
Reviewer 2 Report
The present paper is a systematic review concerning preclinical studies investigating the efficacy of repair with cell therapy of birth asphyxia-induced hypoxic-ischemic encephalopathy.
This is a readable very detailed paper concerning the use of cell therapy in HIE. At the same time the level of detail makes the paper more difficult to interprete the merits of cell therapy for clinical use.
-The Introduction is lengthy and discusses actually the whole textbook with respect to HIE. As the title suggests they should focus on repair/neuroregeneration after a briefintroduction of the pathogenesis and the nowadays therapeutic possibilities of neuroprotective treatment to reduce/prevent birth asphyxia-related brain damage. They should briefly mention the add-on pharmacological therapies for HT investigated and in some instances already used or at least add a reference concerning this issue. It should also be informative for the reader to start with a brief summary of the few clinical studies already provided with autologous or allogenic Cell therapy for neonatal brain damage in a broader sense (UC-derived MSCs or bone marrow-derived MSCs). Only Cotten et al are mentioned here but especially in South Korea there is a strong research group concerning this issue.
-In general the mechanism of the “repairing” effects of UC or bone marrow (M)SC is implicitly stated (often repeatedly) but the paper should benefit to explicitly discussed in a
Separate section, may be a figure can be very instructive (there are several very clear reviews concerning this issue which explain very well the paracrinic and other trophic/ metabolic effects of MSCs as the proposed main action with respect to repair and proliferation of the endogenous NSCs).
-They iterate repeatedly that umbilical derived (mesenchymal) stem cells are more effective then 3rdparty MSCs (bone Marrow) and have a strong opinion about this issue. As far as these reviewers know this issue is based on especially theoretical grounds and in vitro studies, but no real evidence is available in preclinical studies comparing UC-MSCs and Bone Marrow-derived with regard to their repairing effect in the neonatal brain. They should tuning down their statements here a little bit or provide evidence that this is the case. Moreover, important is to realize that “off the self” donor cells are always available and lack of (autologous) UC-derived SCs may be an issue (loss of placenta etc), as is the working-up (culturing) of these cells which cost time. Earlier preclinical studies showed that the efficacy of MSCs after an ischemic-hypoxic blast is up to 10 days, and the discussion what the most optimal point of time after birth asphyxia-related HIE is not at all at this moment. Also the dosing aspect is important: earlier preclinical studies showed that sustained positive effects of MSC-induced repair of the developing brain are dependent of dosing. A sustained repairing effect in mice pups,f.i, after an Vanucci-Rice infarction, was only seen after a half million MSCs. This issue is important when using buffy coats from umblical blood, in which the number of effective SCs is much, much lower (Cotton studies in HIE neonates).
-Some issues related with the described methods:
Methods to optimize stem cells before using it as a therapy (preconditioning) are also an important issue , at least to mention.
A practical issue is to briefly discuss whether or not there is a difference in potency and/or side effects between autologous and allogenic transplantation. IN this review the autologous vs allogenic transplantation is not explicitly discussed.
-I miss a more sorrow discussion (and referencing) about the safety of MSC-transplantation, also vs other SCs “types” (NSCs, engineered SCs, etc). (important ref: Lalu MM et al, PLoS One 2012;7).
-Finally, the summary of ways of administration seems not complete (may be I overlooked this?): Nasal administration of MSCs seems at least as effective as the iv or other routes and it has been shown that they go directly to the brain. I don’t find this back in this review. Actually important clinical safety studies in the newborn patient, running at this moment, use this as the preferred route. An important advantage may be that the cells are not trapped in other (prevoiously Hypxic-ischemic) organ systems and do not reach the brain.
In conclusion, very readable and detailed systemic review with important information. We feel that the above mentioned issues brought up may further improve the paper.
Author Response
"Please see the attachment"

Round 2
Reviewer 2 Report
I have nofurther comments or questions